

# Mehran *vs.* Mehran2 pre-procedure: which score better predicts risk of contrast-induced acute kidney injury in patients with acute coronary syndrome?

Matheus Santos Moitinho[1], Dulce Barbosa[1], Attilio Galhardo[2], Adriano Caixeta[3], Eduesley Santana-Santos[4], Maximina Cunha[1], Beatriz Santana Prado[5,6] and Cassiane Dezoti da Fonseca[1,4,6]

[1] Paulista Nursing School, Federal University of Sao Paulo, Sao Paulo, Brazil
[2] Quebec Heart and Lung Institute, Laval University, Quebec City, Quebec, Canada
[3] Paulista Medical School, Federal University of Sao Paulo, Sao Paulo, Sao Paulo, Brazil
[4] Nursing Post Graduate Program, Federal University of Sergipe, Sao Cristovao, Brazil
[5] Clinical Research Department, Hospital Sírio-Libanês, São Paulo, São Paulo, Brazil
[6] Department of Pathology, Paulista Medical School, Federal University of Sao Paulo, Sao Paulo, SP, Brazil

## ABSTRACT

**Background**. Contrast-induced acute kidney injury (CI-AKI) is a significant concern during percutaneous coronary intervention (PCI) procedures. The novel Mehran 2 pre-procedural risk score, an updated version of the original Mehran score, shows promise as a predictive tool. However, its effectiveness specifically in acute coronary syndrome (ACS) patients requires further investigation. This study aims to evaluate the performance of Mehran 2 pre-procedure risk score compared to original score in predicting CI-AKI risk in acute coronary syndrome patients undergoing PCI.

**Material and Methods**. A prospective cohort study was conducted with patients with ACS undergoing PCI, who were followed up for 90 days (December 2019–February 2021). The Mehran 2 CI-AKI risk score with pre-procedure data was compared with the original Mehran score. Receiver operating characteristic (ROC) curve and area under the ROC curve (AUC-ROC) were used to evaluate the discriminative capacity.

**Results**. 192 patients were analyzed and 33% ($n = 64$) developed CI-AKI. CI-AKI outcome was associated with advanced age, arterial hypertension, chronic kidney disease, troponin T, hemodynamic instability, serum hemoglobin, serum creatinine, and higher both Mehran scores. Both scores demonstrated good agreement. The original Mehran score demonstrated superior CI-AKI stratification with higher sensitivity (85.94%) and specificity (60.16%) compared to the Mehran 2 pre-procedural score (sensitivity 50%, specificity 75%). Significant differences were observed in the discriminative performance between both scores.

**Conclusion**. Sociodemographic, clinical, and laboratory variables were associated with CI-AKI. The original Mehran score demonstrated more consistent discriminative capacity for predicting CI-AKI risk in ACS patients undergoing PCI compared to the Mehran 2 pre-procedural score.

Corresponding author
Cassiane Dezoti da Fonseca,
cassiane.dezoti@unifesp.br

## INTRODUCTION

Cardiovascular diseases (CVD) represent a substantial healthcare challenge due to their high prevalence, multifactorial etiology, and complex treatment strategies (*Précoma et al., 2019*). Among the most prevalent forms of CVD is coronary artery disease (CAD) (*Précoma et al., 2019*; *Nicolau et al., 2021*). The introduction of minimally invasive cardiac interventions like percutaneous coronary intervention (PCI) has revolutionized CAD treatment, particularly in cases of acute coronary syndrome (ACS) with ST-segment elevation (*Nicolau et al., 2021*). However, PCI is associated with the risk of contrast-induced acute kidney injury (CI-AKI), a serious complication associated with the use of contrast agents.

The original Mehran score, developed in 2004 (*Mehran et al., 2004*), is widely used to predict CI-AKI risk post-PCI. However, it relies on post-procedure variables, limiting its utility in urgent settings where early risk assessment is crucial. Additionally, its original development cohort excluded ACS patients, raising concerns about its applicability in this high-risk population. In response to these limitations, *Mehran et al. (2021)* introduced a new score. For this new scoring system, independent predictors of AKI-IC were adopted in two models: Model 1 (pre-procedural), relies solely on pre-procedure variables, whereas Model 2 (post-procedural), incorporating additional variables obtained after PCI. The development and validation of this new score included ACS patients, increasing its potential applicability in high-risk populations.

ACS patients represent a high-risk subgroup within the PCI population, characterized by greater hemodynamic instability, urgent treatment requirements, and a higher burden of comorbidities. These factors not only predispose them to CI-AKI but also limit the time available for renal risk stratification and pre-procedural optimization (*Yang et al., 2018*). Unlike elective PCI patients, ACS patients often require immediate intervention, restricting opportunities for renal function assessment or preventive hydration strategies. Despite this, the original Mehran score, was developed and validated in non-ACS populations, raising concerns about their applicability in this clinically distinct group.

Recent studies have demonstrated a strong association between CI-AKI and poor clinical outcomes in ACS patients undergoing PCI, reinforcing the need for early risk stratification in this population. A meta-analysis found that CI-AKI more than tripled the risk of all-cause mortality in ACS patients (RR = 3.16; 95% CI [2.52–3.97]) (*Yang et al., 2018*). Given the urgent nature of ACS management and the high incidence of CI-AKI in this subgroup, an effective pre-procedural risk stratification tool is essential for guiding preventive strategies and optimizing outcomes. The Mehran 2 pre-procedural score presents an opportunity to improve early CI-AKI risk assessment in ACS patients, yet its predictive performance remains unclear in this specific population, warranting further investigation (*Blanco et al., 2021*; *Mehran et al., 2021*; *Guo et al., 2022*).

Thus, this study aims to evaluate the performance of the Mehran 2 pre-procedure risk score compared to the original score in predicting CI-AKI risk in ACS patients undergoing PCI.

## MATERIALS & METHODS

This study utilized a longitudinal prospective cohort including patients diagnosed with ACS undergoing PCI at a hospital in Brazil. The study was approved by the ethics and research committee under Comitê de Ética em Pesquisa da Universidade Federal de São Paulo (CEP-UNIFESP) opinion no. 3.763.447, patients were recruited from December 2019 to February 2021. Data collection comprised patient interviews, the acquisition of informed consent via signed forms, and medical record analysis. Outcomes were assessed 30 days after the PCI.

The inclusion criteria specified patients aged 18 years or older who were diagnosed with ACS, underwent PCI within seven days of the acute event, remained hospitalized for more than 48 h after PCI, and provided a signed informed consent. The exclusion criteria included individuals without initial or post-PCI serum creatinine values, those transferred between hospitals within 48 h, or those with incomplete medical records, which hindered the assessment of both scores.

GFR was estimated using the 2021 CKD-EPI equation (*Inker et al., 2021*). CI-AKI was defined based on KDIGO criteria as an increase in serum creatinine (Cr) by $\geq 0.5$ mg/dL or a relative increase of $\geq 25\%$ from the baseline within 48 to 72 h post—intervention (*KDIGO, 2012*). CKD was also defined according to KDIGO criteria (*KDIGO, 2012*).

Data collection included the application of both the 2004 original Mehran score and the 2021 Mehran 2 score with pre-procedure variables (Model 1) (*Mehran et al., 2021*). The original Mehran, developed by *Mehran et al. (2004)* incorporates patient-related variables such as age > 75 years, diabetes, heart failure (class III or IV), hypotension, anemia, and chronic kidney disease, alongside procedural factors like intra-aortic balloon use and infused contrast volume. In contrast, *Mehran et al. (2021)* introduced a new score with two models (Model 1 and Model 2) for predicting CI-AKI, with Model 1 excluding procedural variables and focusing on ACS presence, renal function, left ventricular ejection fraction (LVEF), diabetes, serum hemoglobin, glucose, heart failure, and age > 75 years.

This cohort was derived as a secondary objective from a larger study initially designed to evaluate mortality outcomes in the same population, for which a post-hoc power analysis was conducted. Based on the meta-analysis by *Pickering, Blunt & Than (2018)*, the reference incidence of mortality at 30 days of follow-up was 15.0% in patients with CI-AKI and 2.0% in those without CI-AKI. Assuming a null hypothesis ($H_0$: Pr(Y=1|X=1) = 0.02), a significance level ($\alpha = 0.05$), and a total sample size of 192 patients, the achieved statistical power for detecting this association was 0.98. The large sample z-test method, as proposed by *Demidenko (2007)*, was employed for this calculation, incorporating variance correlation adjustments to enhance the reliability of the estimate. The high statistical power obtained supports the robustness of this cohort in detecting clinically meaningful associations between CI-AKI and mortality.

Data were stored using the REDCap (*Harris et al., 2009*) platform for data collection and subsequently entered into a Microsoft$^{\circledR}$ Excel-2010 electronic spreadsheet for analysis.

Normality and homogeneity of quantitative variables were assessed using Shapiro–Wilk and Levene's tests. Quantitative variables were reported using either the mean or median

(Md), along with the standard deviation (±) or interquartile range (IQR) (p25, p75). Categorical variables were presented through both absolute and relative frequencies (%). Continuous variables were compared between groups, delineated by the development of CI-AKI, using either the independent samples $t$-test (t) for normally distributed data or the Mann–Whitney U test (U) for non-normally distributed data. In the presence of heterogeneity of variances, the Welch correction was applied to the $t$-test. Categorical variables were analyzed using Pearson's chi-square test ($\chi 2$) with Yates correction or Fisher's exact test.

The reliability of the original Mehran and Mehran 2 pre-procedural tools was evaluated using intraclass correlation coefficients (ICC), and model quality was assessed through the Cronbach's alpha ($\alpha$) index. Receiver operating characteristic (ROC) curves were constructed to gauge the predictive performance of the models for CI-AKI, with the area under the ROC curve (AUC-ROC) calculated. The Delong test was used to compare AUC-ROCs between the two scores, and diagnostic performance metrics such as sensitivity and specificity were derived from the ROC curves to determine potential cutoff values.

The association between Mehran tools with the incidence of CI-AKI were explored using a Poisson regression model with robust variance and a log-link function, with estimators interpreted as relative risks (RR) (*Zou, 2004*). Overdispersion was tested, and the Poisson model was compared to a negative binomial model using the Akaike information criterion (AIC). To assess the predictive performance of the Mehran original and Mehran 2 pre-procedural scores, net reclassification improvement (NRI) and integrated discrimination improvement (IDI) were calculated using categorical and continuous reclassification methods, with cutoff points based on risk prediction quartiles. All statistical estimates were given a 95% confidence interval (CI 95%), and significance was established at $p < 0.05$. To assess the magnitude of the observed effects, effect size estimators such as Cohen's d (d), Cramer's V, and point-biserial correlation (rb) were employed. All statistical analyses were conducted using Jamovi 2.2.5® software ("*The jamovi project (2022)*. jamovi (Version 2.3) (Computer Software). Retrieved from https://www.jamovi.org").

## RESULTS

A total of 192 patients were included in this study, revealing a 33% incidence rate of CI-AKI ($n = 64$). CI-AKI was significantly associated with age. CI-AKI patients had a median age of 65 years, compared to 61 years in non-CI-AKI patients, with a moderate effect size ($p < 0.012$; d: −0.528, respectively) (Table 1).

Comorbidity analysis indicated that hypertension was significantly more prevalent in the CI-AKI group (77.8% *vs.* 61.3%; $p = 0.031$; Cramer's V: 0.165). Similarly, chronic kidney disease (CKD) was considerably more frequent among patients with CI-AKI (18.8% *vs.* 3.9%; $p < 0.001$; Cramer's V: 0.246) (Table 1).

While the distribution of ACS diagnoses did not statistically impact CI-AKI outcomes ($P = 0.481$), other factors such as high-sensitivity Troponin T levels at admission were significantly associated. Specifically, median Troponin T levels were higher in the CI-AKI group (329 ng/L) compared to those without (92 ng/L), with a significant difference ($p = 0.035$; rb: 0.201).

**Table 1  Baseline sociodemographic and clinical characteristics of patients with acute coronary syndrome undergoing percutaneous coronary intervention, stratified by contrast-induced acute kidney injury status.** Baseline sociodemographic and clinical characteristics of patients undergoing PCI, grouped by CI-AKI status. Data are presented as n (%), mean ± SD, or median (IQR). Statistical tests: Pearson chi-square ($\chi^2$) for categorical variables, Student's t-test (t) for normally distributed continuous variables, and Mann-Whitney U test (U) for non-normal distributions. $p < 0.05$ considered significant.

| Clinical-sociodemographic and PCI-related | Total $n = 192$ | CI-AKI | | Statistical test |
|---|---|---|---|---|
| | | No ($n = 128$) | Yes ($n = 64$) | |
| **Male**$_{n(\%)}$ | 137/192 (71.4%) | 95/128 (74.2%) | 42/64 (65.6%) | $\chi^2_{(1)} = 1.54, P = 0.21$[1] |
| **Age**$_{Mean(\pm)}$ | 61.9 (10.6) | 60.1 (10.8) | 65.6 (9.2) | $t_{(190)} = -3.45, P < 0.01$[2] |
| **BMI**$_{Md(IQR)}$ | 26.8 (24.4–29.8) | 27 (24.6–29.8) | 26.4 (23.8–29.7) | $U = 3.826, P = 0.56$[4] |
| **Smoking**$_{n(\%)}$ | | | | |
| Smoker /Ex Smoker | 130/192 (67.7%) | 91/128 (71.1%) | 39/64 (60.9%) | $\chi^2_{(1)} = 2.01, P = 0.16$[1] |
| No Smoker | 62/192 (32.3%) | 37/128 (28.9%) | 25/64 (39.1%) | |
| **SAH**$_{n(\%)}$ | 125/187 (66.8%) | 76/124 (61.3%) | 49/63 (77.8%) | $\chi^2_{(1)} = 4.41, P = 0.03$[1] |
| **CKD**$_{n(\%)}$ | 17/192 (8.9%) | 5/128 (3.9%) | 12/64 (18.8%) | $\chi^2_{(1)} = 11.6, P < 0.01$[1] |
| **DM**$_{n(\%)}$ | 58/192 (30.2%) | 34/128 (26.6%) | 24/64 (37.5%) | $\chi^2_{(1)} = 2.42, P = 0.12$[1] |
| **Treatment for DM**$_{n(\%)}$ | | | | |
| Oral medication | 35/192 (18.2%) | 19/128 (14.8%) | 16/64 (25%) | |
| Insulin therapy | 15/192 (7.8%) | 8/128 (6.3%) | 7/64 (10.9%) | $\chi^2_{(3)} = 5.91, P = 0.12$[2] |
| Non-adherent | 8/192 (4.2%) | 7/128 (5.5%) | 1/64 (1.6%) | |
| Without DM | 134/192 (69.8%) | 94/128 (73.4%) | 40/64 (62.5%) | |
| **Dyslipidemia**$_{n(\%)}$ | 87/189 (46%) | 56/127 (44.1%) | 31/62 (50%) | $\chi^2_{(1)} = 0.58, P = 0.44$[1] |
| **HF**$_{n(\%)}$ | 21/192 (10.9%) | 11/128 (8.6%) | 10/64 (15.6%) | $\chi^2_{(1)} = 2.17, P = 0.14$[1] |
| **ACS clinical presentation**$_{n(\%)}$ | | | | |
| Unstable angina | 31/192 (16.1%) | 23/128 (18.0%) | 8/64 (12.5%) | |
| STEMI | 98/192 (51.0%) | 66/128 (51.6%) | 32/64 (50.0%) | $\chi^2_{(2)} = 1.45, P = 0.48$[1] |
| NSTEMI | 63/192 (32.8%) | 39/128 (30.5%) | 24/64 (37.5%) | |
| **Troponin T Admission (ng/L)**$_{Md(IQR)}$ | 136 (35–1137) | 92 (31–1087) | 329 (47–2431) | $U = 2.875, P = 0.03$[4] |
| **KILLIP**$_{n(\%)}$ | | | | |
| I–II | 181/192 (94.3%) | 122/128 (95.3%) | 50/64 (92.2%) | $\chi^2_{(1)} = 0.77, P = 0.39$[1] |
| III–IV | 11/192 (5.7%) | 6/128 (4.7%) | 9/64 (7.8%) | |
| **Vasoactive drugs**$_{n(\%)}$ | 21/191 (11.0%) | 8/127 (6.3%) | 13/64 (20.3%) | $\chi^2_{(1)} = 8.54, P < 0.01$[1] |
| **Thrombolysis**$_{n(\%)}$ | 25/191 (13.1%) | 20/128 (15.6%) | 5/64 (7.9%) | $\chi^2_{(1)} = 2.19, P = 0.14$[1] |
| **Cardiorespiratory arrest**$_{n(\%)}$ | 21/192 (10.9%) | 6/128 (4.7%) | 15/64 (23.4%) | $\chi^2_{(1)} = 15.40, P < 0.01$[1] |
| **Hb Admission (mg/dl)**$_{Md(IQR)}$ | 14.2 (12.6–15.8) | 14.5 (13.0–15.9) | 13.4 (11.9–14.9) | $U = 2.670, P < 0.01$[4] |
| **GFR Admission (mL/min)**$_{Md(IQR)}$ | 81.0 (56.0–101) | 65.2 (54.9–77.7) | 60.6 (37.8–87.5) | $U = 3.963, P = 0.724$ |
| **Cr admission (mg/dl)**$_{Md(IQR)}$ | 1.0 (0.82–1.2) | 1.0 (0.9–1.2) | 1.0 (0.8–1.5) | $U = 3.982, P = 0.75$[4] |
| **Contrast volume (ml)**$_{Md(IQR)}$ | 150 (120–200) | 150 (130–200) | 150 (112–190) | $U = 3.755, P = 0.75$[4] |
| **Number of stents**$_{Md(IQR)}$ | 1 (1–2) | 1 (1–2) | 1 (1–2) | $U = 3.505, P = 0.404$ |
| **PCI duration**$_{Md(IQR)}$ | 50.0 (35.0–65.8) | 48.0 (34.9–65.0) | 58.5 (35.4–70.0) | $U = 3.370, P = 0.23$[4] |
| **Mehran original score**$_{Md(IQR)}$ | 5.0 (1.0–9.0) | 4.0 (1.0–6.6) | 7.5 (5.0–12.6) | $U = 2.232, P < 0.01$[4] |
| **Categories original Mehran**$_{n(\%)}$ | | | | |
| Low risk | 106/192 (55.2%) | 84/128 (65.6%) | 22/64 (34.4%) | |

**Table 1** (*continued*)

| Clinical-sociodemographic and PCI-related | Total $n = 192$ | CI-AKI | | Statistical test |
|---|---|---|---|---|
| | | No ($n = 128$) | Yes ($n = 64$) | |
| Moderate risk | 51/192 (26.6%) | 30/128 (23.4%) | 21/64 (32.8%) | $\chi^2(3)$=20.18, $P < 0.01$[1] |
| High risk | 23/192 (12.0%) | 9/128 (7.0%) | 14/64 (21.9%) | |
| Very high risk | 12/192 (6.2%) | 5/128 (3.9%) | 7/64 (10.9%) | |
| **Mehran 2 pre-procedural**$_{Md(IQR)}$ | 8.0 (5.0–9.0) | 8.0 (4.0–8.6) | 8.5 (6.0–10.0) | $U = 2.958$, $P < 0.01$[4] |
| **Categories Mehran 2 pre-procedural**$_{n(\%)}$ | | | | |
| Low risk | 13/192 (6.8%) | 12/128 (9.38%) | 1/64 (1.56%) | |
| Moderate risk | 115/192 (59.9%) | 84/128 (65.63%) | 31/64 (48.44%) | $\chi^2(3)$=16.52, $P < 0.01$[3] |
| High risk | 56/192 (29.2%) | 30/128 (23.44%) | 26/64 (40.63%) | |
| Very high risk | 8/192 (4.2%) | 2/128 (1.56%) | 6/64 (9.38%) | |

**Notes.**
CI-AKI, Contrast-Induced Acute Kidney Injury; BMI, Body Mass Index; SAH, Systemic Arterial Hypertension; CKD, Chronic Kidney Disease; DM, Diabetes Mellitus; HF, Heart Failure; ACS, Acute Coronary Syndrome; STEMI- ST, Elevation Myocardial Infarction; NSTEMI, Non-ST-Elevation Myocardial Infarction; Hb, Serum Hemoglobin; GFR, Glomerular Filtration Rate; Cr, Serum Creatinine; PCI, Percutaneous Coronary Intervention.
[1] Pearson Chi-square.
[2] t de student.
[3] Chi-square with Yates correction.
[4] Mann–Whitney.

Vasoactive drug use and cardiopulmonary arrest were significantly associated with CI-AKI ($p < 0.01$; Cramer's V: 0.211 and 0.283, respectively) (Table 1).

CI-AKI patients had lower median hemoglobin at admission, 24, and 48 h, with a discrete but statistically significant correlation to CI-AKI outcomes ($p = 0.005$, 0.007, 0.013; rb: 0.258, 0.252, 0.238, respectively). However, no significant differences were observed in admission creatinine values ($p = 0.750$) (Table 1).

The original Mehran score demonstrated a significant difference in the context of CI-AKI ($p < 0.001$, rb:0.455). The statistical analysis of Mehran 2 pre-procedural also revealed significant results, although with a smaller effect size ($p = 0.001$, rb:0.278). In terms of score categories, the highest values of the moderate, high, and very high-risk categories of the original Mehran score are significantly associated with the incidence of CI-AKI ($p < 0.001$; Cramer's V:0.342), as well as the highest values of the high and very high-risk categories of the Mehran 2 pre-procedural score for CI-AKI ($p < 0.001$; Cramer's V:0.293).

To assess the reliability and consistency of the scores, the degree of agreement between the scores obtained by different tools was examined through ICC analysis. It was observed that the measurements showed good agreement (ICC: 0.603, $\alpha$: 0.620, $p < 0.001$).

For the Mehran score, three cut-off points (three, four, and five) were highlighted, where sensitivity ranged from 76.56% to 85.94%, specificity ranged from 46.88% to 60.16%, the Youden index (evaluating the balance between sensitivity and specificity) varied from 0.328 to 0.367, and the AUC-ROC was 0.728. Regarding the Mehran 2 pre-procedural score, a single cut-off point of eight points was observed, with a sensitivity of 50% and specificity of 75%. The AUC-ROC for the Mehran 2 pre-procedural score is 0.639, with a Youden index of 0.250 (Table 2).

**Table 2 Diagnostic performance metrics of the original Mehran score and Mehran 2 pre-procedural score at different cut-off points in patients with acute coronary syndrome undergoing percutaneous coronary intervention.** Diagnostic metrics of the original Mehran score and Mehran 2 pre-procedural score for CI-AKI risk prediction. Metrics include sensitivity, specificity, positive predictive value (PPV), negative predictive value (NPV), Youden's Index, and AUC-ROC. Higher AUC-ROC indicates better discriminative ability.

| Cut-off point | Sensitivity (%) | Specificity (%) | PPV (%) | NPV (%) | Youden's index | AUC-ROC | Metric score |
|---|---|---|---|---|---|---|---|
| **Mehran riginal** | | | | | | | |
| 3 | 85.94% | 46.88% | 44.72% | 86.96% | 0.3281 | 0.73 | 1.33 |
| 4 | 84.38% | 49.22% | 45.38% | 86.30% | 0.3359 | 0.73 | 1.34 |
| 5 | 76.56% | 60.16% | 49% | 83.70% | 0.3672 | 0.73 | 1.37 |
| **Mehran 2 pre-procedural** | | | | | | | |
| 8 | 50% | 75% | 50% | 75% | 0.2500 | 0.64 | 1.25 |

**Notes.**
PPV, Positive Predictive Value; NPV, Negative Predictive Value; AUC-ROC, Area under the Receiver Operating Characteristic Curve.

Moreover, the analysis of the difference between the AUC-ROCs reveals a statistically significant difference in the performance of the original Mehran score and the Mehran 2 pre-procedural score ($p = 0.034$). This indicates that the original Mehran score statistically demonstrates a better discriminative capacity compared to the Mehran 2 pre-procedural score, as show in the first figure (Table 2 and Fig. 1).

It was observed that the diagnostic performance metrics classified by severity level generated by both scales show that the Mehran 2 pre-procedural score achieved better sensitivity with the Moderate risk classification (48.44%) and better specificity when the risk was Very High (98.44%). Notably, the Moderate category demonstrates a relevant metric for detecting CI-AKI among those who truly had renal involvement, while the Very High classification exhibits substantial accuracy in detecting true negatives among those without CI-AKI (Table 3).

Evaluating subgroups of the different ACS diagnoses, the analysis revealed differences in the performance of the Mehran 2 pre-procedural and original Mehran scores in predicting CI-AKI across different ACS subgroups. In non-ST-elevation myocardial infarction (NSTEMI) patients, the original Mehran score demonstrated higher sensitivity (91.67%) and negative predictive value (NPV = 89.47%), suggesting better reliability in ruling out the disease. Conversely, the Mehran 2 pre-procedural score achieved a more balanced sensitivity (66.67%) and specificity (66.67%), indicating a moderate ability to identify both affected and unaffected patients. In the unstable angina subgroup, the Mehran 2 pre-procedural score exhibited superior sensitivity (87.5%) and NPV (92.31%), favoring its use in exclusion, while the original Mehran score showed greater specificity (73.91%) and positive predictive value (PPV = 45.45%), enhancing its confirmatory capability. Among ST-elevation myocardial infarction (STEMI) patients, both scores performed similarly, with the original Mehran score demonstrating slightly higher specificity (63.64%) and PPV (51.02%), while the Mehran 2 pre-procedural score maintained higher sensitivity (81.25%) and NPV (85.71%) indicating better exclusion power (Table 4).

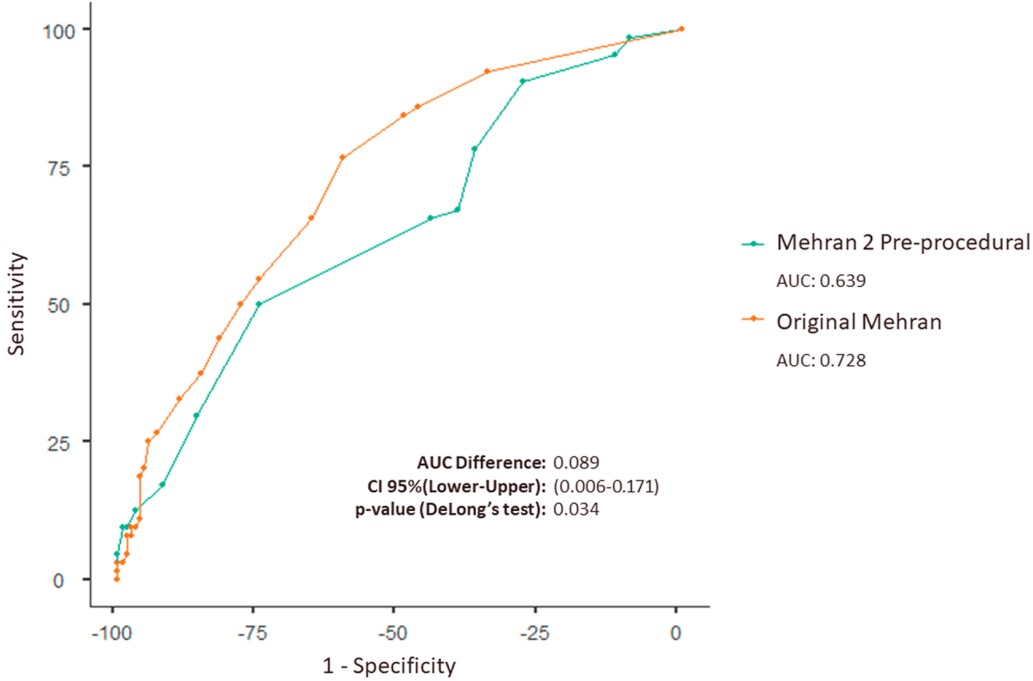

**Figure 1** **Combined ROC curve demonstrating diagnostic performance for the original Mehran score and Mehran 2 pre-procedural in patients with ACS undergoing PCI.** ROC curve comparing the original Mehran (AUC: 0.728) and Mehran 2 pre-procedural (AUC: 0.639) scores for predicting outcomes in ACS patients undergoing PCI. The AUC difference is 0.089 (95% CI [0.006 to 0.171], $p = 0.034$), showing superior accuracy of the original Mehran score.

The analysis of the relationship between the Mehran tools and the incidence of CI-AKI showed no overdispersion in the Poisson model with original Mehran and Mehran 2 pre-procedural ($z = -5.002$, $p = 1.000$, dispersion $= 0.667$ and $z = -5.105$, $p = 1.000$, dispersion $= 0.667$). Additionally, the Poisson model demonstrated better values of AIC than negative binomial model for both instruments (258 *vs.* 260 and 265.6 *vs.* 267.63).

In the analysis of predictive power using the Poisson model for the incidence of CI-AKI based on the original Mehran score, it is observed that the incidence of CI-AKI ($n = 128/33\%$) increases approximately 7% for each one-unit increase in the original Mehran score (RR: 1.077; 95% CI [1.051 to 1.103]). Regarding the model with Mehran 2 pre-procedural, it is noted that the fit quality of the model to the data was worse compared to the original (AIC: 265.6 *vs.* AIC: 258) and demonstrated that for each one-unit increase in the Mehran 2 pre-procedural score, there was an 11% increase in the rates of CI-AKI (RR: 1.116; 95% CI [1.054 to 1.183]).

Reclassification analysis showed that the Mehran 2 pre-procedural score did not significantly improve risk reclassification compared to the original Mehran (Categorical NRI $= -0.267$; 95% CI [$-0.608$ to $-0.075$]; $p = 0.126$). However, the Continuous NRI was negative and statistically significant ($-0.453$; 95% CI [$-0.746$ to $-0.161$]; $p = 0.002$), indicating a worsening of continuous risk reclassification with Mehran 2. Similarly, the IDI was negative and significant ($-0.071$; 95% CI [$-0.115$ to $-0.027$]; $p = 0.001$), suggesting a

**Table 3** **Diagnostic performance of the original Mehran score and Mehran 2 pre-procedural score by RisR category in patients with acute coronary syndrome undergoing percutaneous coronary intervention.** Diagnostic metrics of the original Mehran score and Mehran 2 pre-procedural score across risk categories (Low, Moderate, High, Very High) in predicting CI-AKI. Includes sensitivity, specificity, accuracy, PPV, NPV, post-test probabilities, and likelihood ratios.

| | Mehran original | | | | Mehran 2 pre-procedural | | | |
|---|---|---|---|---|---|---|---|---|
| | Low | Moderate | High | Very high | Low | Moderate | High | Very high |
| Sensitivity[a] | 34.38% | 32.81% | 21.88% | 10.94% | 1.56% | 48.44% | 40.63% | 9.38% |
| Especificidade[b] | 34.38% | 76.56% | 92.97% | 96.09% | 90.63% | 34.38% | 76.56% | 98.44% |
| Accuracy[d] | 34.38% | 61.98% | 69.27% | 67.71% | 60.94% | 39.06% | 64.58% | 68.75% |
| Incidence[e] | 33.33% | 33.33% | 33.33% | 33.33% | 33.33% | 33.33% | 33.33% | 33.33% |
| PPV[f] | 20.75% | 41.18% | 60.87% | 58.33% | 7.69% | 26.96% | 46.43% | 75.00% |
| NPV[g] | 51.16% | 69.50% | 70.41% | 68.33% | 64.80% | 57.14% | 72.06% | 68.48% |
| Post-test Disease Probability[h] | 20.75% | 41.18% | 60.87% | 58.33 | 7.69% | 26.96% | 46.43% | 75.00% |
| Post-test Healthy Probability[i] | 51.16% | 69.50% | 70.41% | 68.33% | 64.80% | 57.14% | 72.06% | 68.48% |
| Positive likelihood ratio | 0.5238 | 1.400 | 3.111 | 2.800 | 0.1667 | 0.7381 | 1.733 | 6.000 |
| Negative likelihood ratio | 1.909 | 0.8776 | 0.8403 | 0.9268 | 1.086 | 1.500 | 0.7755 | 0.9206 |

Notes.
[a]Sensitivity (True Positives among Diseased).
[b]Specificity (True Negatives among Healthy).
[d]Accuracy (Rate of True Test Results).
[e]Incidence of the disease in this population.
[f]Positive Predictive Value (Probability of having the disease after a positive test using this experimental population).
[g]Negative Predictive Value (Probability of being healthy after a negative test in this experimental population).
[h]Post-test Probability of having the disease (Probability of having the disease after a positive test using known population incidence).
[i]Post-test Probability of being healthy (Probability of being healthy after a negative test using known population prevalence).

**Table 4** **Diagnostic performance of the original Mehran score and the Mehran 2 pre-procedural score in predicting CI-AKI across acute coronary syndrome subgroups in patients undergoing percutaneous coronary intervention.** Diagnostic performance of the original Mehran score and the Mehran 2 pre-procedural score in predicting CI-AKI across acute coronary syndrome subgroups. Metrics include sensitivity, specificity, accuracy, positive predictive value (PPV), negative predictive value (NPV).

| Escore | Unstable angina | | NSTEMI | | STEMI | |
|---|---|---|---|---|---|---|
| | Mehran 2 pre-procedural | Original Mehran | Mehran 2 pre-procedural | Original Mehran | Mehran 2 pre-procedural | Original Mehran |
| Cut-off Point | 2.50 | 5.50 | 5.50 | 6.50 | 8.50 | 4.50 |
| Sensitivity[a] | 87.50% | 62.50% | 66.67% | 91.67% | 81.25% | 78.13% |
| Especificidade[b] | 52.17% | 73.91% | 66.67% | 43.59% | 54.55% | 63.64% |
| Accuracy[c] | 61.29% | 70.97% | 66.67% | 61.90% | 63.27% | 68.37% |
| PPV[d] | 38.89% | 45.45% | 55.17% | 50.00% | 46.43% | 51.02% |
| NPV[e] | 92.31% | 85.00% | 76.47% | 89.47% | 85.71% | 85.71% |

Notes.
[a]Sensitivity (True Positives among Diseased).
[b]Specificity (True Negatives among Healthy).
[c]Accuracy (Rate of True Test Results).
[d]Positive Predictive Value (Probability of having the disease after a positive test using this experimental population).
[e]Negative Predictive Value (Probability of being healthy after a negative test in this experimental population).

lower ability of Mehran 2 to discriminate between patients with and without AKI compared to the original Mehran. These findings indicate that the Mehran 2 pre-procedural score does not improve risk stratification and may have lower predictive performance for AKI following percutaneous coronary intervention than original Mehran.

## DISCUSSION

This study evaluates the effectiveness of the Mehran 2 pre-procedural score in predicting CI-AKI in a high-risk population. The replication of this tool in variable samples and populations, contrasting with previous North American and Asian samples used in development, validation, and performance studies, represents a point of significant relevance for this investigation (*Mehran et al., 2021*; *Guo et al., 2022*). It is important to highlight that in this study, the original Mehran score demonstrated superior performance compared to the Mehran 2 pre-procedural score in terms of predictive metrics.

Our findings not only quantify and expand the understanding of the performance of these two risk scores, but also underscores the importance of assessing the applicability of scientific tools across different populations. Evaluating the performance of clinical tools such as the original Mehran and Mehran 2 pre-procedural scores with pre-procedure data in specific contexts is critical for validating their effectiveness and utility in diverse clinical settings. The ability to replicate scientific findings, particularly in typical clinical scenarios different from the original tool development settings, contributes to a more comprehensive and robust understanding of the implementation of these risk assessment tools. This helps ensure these instruments are widely used, reliable, and clinically useful in real-world practice.

The findings of this study are significant, as they contrast with existing evidence regarding the performance of the Mehran 2 pre-procedural score, compared to the original Mehran score. Previous research, including its development ($n = 14.616$) and validation ($n = 5.606$) cohorts, as well as subsequent studies with an Asian population ($n = 2.487$), consistently demonstrated the equivalence of the Mehran 2 pre-procedural score to the original Mehran score, showing high predictive power (ROC 0.836) and excellent data fit (Mehran 2 pre-procedural to the model $\chi 26 = 5.320$, $P = 0.503$) (*Guo et al., 2022*).

While previous studies, particularly those conducted in North American and Asian populations (*Blanco et al., 2021*; *Mehran et al., 2021*; *Guo et al., 2022*), have demonstrated equivalence between the two scores, our findings revealed the superior predictive capacity of the original Mehran score in ACS patients.

These discrepancies may stem from variations in population characteristics, including ethnicity, comorbidities, and clinical severity. The original Mehran score was validated in a more diverse population, whereas the Mehran 2 pre-procedural score incorporated additional variables specific to ACS, which may have influenced its performance when applied exclusively to high-risk ACS patients. Our study cohort had different characteristics, like higher prevalence of systemic arterial hypertension (SAH), CKD, and hemodynamic instability, which may have impacted the calibration of Mehran 2 pre-procedural predictive ability. These findings underscore the need for further validation in different populations to assess whether regional and demographic variations influence the accuracy of CI-AKI risk stratification tools.

Furthermore, this discrepancy from previous findings could be attributed to intentional sampling bias in our study, focusing solely on patients with acute coronary syndrome (ACS), without stable angina patients. The Mehran 2 pre-procedural score incorporates

additional variables related to ACS, scoring the type of clinical presentation. In our study, the absence of patients with stable angina resulted in a skewed distribution of baseline scores for Mehran 2 pre-procedural, with values of two, four, or eight points depending on ACS severity. This imbalance amplified score discrepancies, complicating a fair comparison with the original Mehran score. The additional ACS variable in Mehran 2 pre-procedural score likely affected accuracy metrics.

The original Mehran score exhibited higher sensitivity (86% at a score of three) for detecting CI-AKI, while the Mehran 2 pre-procedural score demonstrated greater specificity (75% at a score of eight) for ruling out CI-AKI. The Mehran 2 score is particularly useful for identifying low-risk. The Mehran 2 pre-procedural score's superior specificity (75%) makes it particularly useful in preventing unnecessary interventions by accurately identifying patients at low risk for CI-AKI. This is especially relevant in patients with preserved renal function, where excessive hydration may be unnecessary, in resource-limited settings, where targeted nephroprotection is crucial, and in urgent PCI cases, where rapid pre-procedural risk assessment is required. In contrast, the original Mehran score's higher sensitivity (85.94%) enhances early detection of at-risk patients, making it particularly valuable in CKD patients, elective PCI cases requiring pre-procedural optimization, and high-risk ACS patients with hemodynamic instability, where intensive renal protection strategies can be implemented proactively.

Notably, subgroup analysis revealed that Mehran 2 pre-procedural demonstrated greater negative predictive value (NPV) across ACS subtypes, particularly in unstable angina (92%) and STEMI (86%), favoring its role in disease exclusion. In contrast, the original Mehran score exhibited higher positive predictive value (PPV) in these groups (45.45% and 51.02%, respectively), making it more suitable for confirmatory diagnosis.

Given these distinct advantages, a hybrid approach may optimize CI-AKI risk stratification, by using the original Mehran score for high-risk ACS patients requiring aggressive preventive measures and the Mehran 2 pre-procedural score for broader, pre-procedural risk assessment, particularly in settings where reducing overtreatment is a priority. This strategy could balance early detection with intervention precision, improving risk prediction and patient outcomes.

Additionally, by focusing solely on ACS patients, this study provides valuable insights into CI-AKI risk in a high-acuity setting where rapid decision-making is crucial. However, the intentional exclusion of stable angina patients in our study impacts the broader applicability of our findings, as their clinical profile differs significantly from that of ACS patients. Stable angina is commonly managed with elective PCI, allowing for better pre-procedural optimization, including hydration protocols and renal function assessment. However, future research should evaluate the performance of risk stratification tools, such as the Mehran 2 pre-procedural score, in stable angina populations. Expanding predictive models to include both stable and unstable coronary syndromes could enhance their clinical utility, ensuring optimal CI-AKI prevention strategies across all PCI patients.

However, in line with our result, a recent study found that the excluding contrast from the pre-procedural Mehran 2 score resulted in a small but significant loss of discriminatory

capacity compared to the original Mehran score (AUROC 0.73 *versus* 0.74; $P = 0.01$) (*Blanco et al., 2021*).

Regarding the risk classification from the scores, we observed a maximum sensitivity of 34.38% for the original Mehran low-risk category and 48.44% for the Mehran 2 pre-procedural moderate-risk category. It is noteworthy that a prior performance study of these instruments reported a sensitivity of 95.29% for the low-risk group with Mehran 2 pre-procedural, contrasting with our findings (*Blanco et al., 2021*). Notably, the specificity of Mehran 2 pre-procedural for the very high-risk score was 98.44%, surpassing the original Mehran score at 96.09%. These measures align closely with those described in a study analyzing the performance of Mehran 2 pre-procedural, which reported a specificity of 95.44% for the high-risk group (*Blanco et al., 2021*).

In analyzing these results, we emphasize the influence of population size on the accuracy of the confusion matrix in detecting a relatively infrequent outcome, affected by its distribution among risk classifications. Dividing the patients into four distinct risk categories reduces the analytical power for each category and decreases the frequency of the outcome of interest within each category. This may lead to inaccurate estimates, particularly in metrics sensitive to event distribution such as sensitivity and specificity. Therefore, it is plausible that the observed sensitivity for the scores may be underestimated due to population size restrictions.

Despite the superiority demonstrated by the original Mehran score in terms of model accuracy, each unit variation in the Mehran 2 pre-procedural score had a more substantial impact on predicting the incidence of CI-AKI (RR: 1.077 *versus* 1.116, respectively). This analysis suggests that Mehran 2 pre-procedural exhibits heightened sensitivity in predicting CI-AKI, despite slightly lower data adherence. With an AUC-ROC of 0.64, Mehran 2 pre-procedural emerges as an efficient tool for CI-AKI detection, excelling particularly in the precise identification of patients who will not develop the outcome. This capability is crucial in clinical settings where excluding false positives is imperative to avoid unnecessary interventions or undue concerns, for instance, in scenarios involving preventive strategies for CI-AKI, such as intravenous hydration administration in patients with compromised cardiac, pulmonary function, or nephrotic syndrome.

Conversely, the original Mehran score demonstrates superior sensitivity, indicating its effectiveness in identifying patients with CI-AKI. This ability holds great importance in clinical scenarios where early CI-AKI detection is critical for prompt patient treatment and monitoring. However, it is essential to acknowledge that the original Mehran score relies also on post-procedure information obtained after PCI, limiting its applicability in pre-procedural settings, particularly in clinical contexts where early identification is paramount.

Therefore, the availability of pre-procedural Mehran 2 pre-procedural variables prior to PCI renders it valuable for promptly identifying high-risk patients for CI-AKI before the intervention. This facilitates the implementation of prophylactic measures (*Moitinho et al., 2020*), such as isotonic hydration, suspension of nephrotoxic medications, fluid volume control, and scheduled procedure delays (*Brown et al., 2014*). Such straightforward and

cost-effective measures have demonstrated a reduction of up to 20% in CI-AKI incidence, as evidenced by a robust multicenter study in 2014 (*Brown et al., 2014*).

Our study comprehensively assessed the effects and associations of various variables with the occurrence of contrast-induced acute kidney injury (CI-AKI). Notably, advanced age emerged as a significant factor for CI-AKI occurrence, corroborating existing scientific literature that identifies advanced age as a widely recognized risk factor for CI-AKI development (*Pan et al., 2018*; *Wang et al., 2021*). This susceptibility in elderly individuals may stem from physiological impairments, including reduced renal functional reserve and diminished cellular regenerative capacity (*Jufar et al., 2020*). Additionally, the presence of common comorbidities like DM and hypertension further heightens renal vulnerability to contrast exposure, as documented in a Chinese study (*Pan et al., 2018*).

In line with this, our study revealed notable associations between CI-AKI and comorbidities such as SAH and CKD. The associations between CI-AKI and CKD is well-documented in scientific literature, supported by a 2019 review that underscores the increased risk associated with DM and CKD (*Vlachopanos et al., 2019*). These conditions moderate renal injury by exacerbating medullary hypoxia generated by tubular cytotoxicity and are responsible for reducing renal mass and contributing to endothelial dysfunction. Similarly, a robust meta-analysis, summarizing data from over 169.455 patients identified hypertension as a significant factor associated with increased CI-AKI risk ($p = 0.03$) (*Obed et al., 2022*).

These findings reinforce the importance of carefully considering comorbidities such as DM, SAH, and CKD in assessing the risk of CI-AKI in patients undergoing contrast procedures. It is relevant to emphasize that the Mehran scores incorporate DM, baseline glucose, age, and renal function as significant variables (*Mehran et al., 2004*; *Mehran et al., 2021*).

Our findings showed a significant association between elevated levels of admission troponin T and low serum hemoglobin with the incidence of CI-AKI. The rise in troponin T levels may indicate a correlation with the level of myocardial injury, *i.e.*, the extent of cardiac tissue compromise, can reflect in reduced cardiac output and, consequently, may contribute to reduced renal perfusion. This scenario may lead to renal hypoperfusion, contributing to the development of AKI (*Basile, Anderson & Sutton, 2012*). This finding is consistent with previous research in septic patients, where admission troponin T levels predicted AKI and the need for dialysis (*De Almeida Thiengo et al., 2018*).

Although the original Mehran score does not account for variables related to myocardial tissue damage from ACS, the updated Mehran 2 pre-procedural score incorporates clinical ACS presentation, including pre-exam serum hemoglobin levels.

Low serum hemoglobin levels, indicative of anemia, can reduce tissue oxygenation, particularly in the renal medulla. A previous analysis from this cohort found that for every unit decrease in admission serum Hb, the likelihood of CI-AKI increased by 6.5% (*Moitinho et al., 2022*), as well as a study from the USA that demonstrated that for every 3% reduction in baseline hematocrit, the chances of developing the CI-AKI outcome increased (*Nikolsky et al., 2005*). Hypoxia resulting from low Hb concentration can play a substantial role in the progression of renal injury, especially in the context of contrast-induced nephrotoxicity

(*Nikolsky et al., 2005*). The interplay between cardiac dysfunction and inadequate tissue oxygenation is likely a critical component in the pathogenesis of CI-AKI.

It is essential to acknowledge the limitations of our study when interpreting the results. Firstly, the single-center design and limited population size may affect the precision, generalizability, and reproducibility of our findings. Conducting the study at a single institution in Brazil introduces potential selection bias, as patient demographics, clinical practices, and healthcare infrastructure may differ from those in other regions or countries. The limited number of participants may reduce statistical power, affecting the robustness of subgroup analyses and the ability to detect smaller but clinically relevant associations. The restricted population size to patients with ACS limited the possibility of a fair comparison between the scores, compromising the robustness of conclusions related to the equivalence of these assessment tools. To improve external validity, future multicenter studies with larger and more diverse populations are essential. Expanding research to multiple institutions, including centers with varying patient profiles and healthcare protocols, would enhance the applicability of findings and provide a more comprehensive assessment of CI-AKI risk stratification tools, ensuring broader clinical relevance.

## CONCLUSIONS

In summary, our study emphasized the performance of the Mehran 2 pre-procedural score for predicting CI-AKI in patients undergoing PCI, particularly those with ACS. While demonstrating acceptable accuracy, the Mehran 2 score underperformed slightly compared to the original Mehran score in our study population. The significant associations observed between age, comorbidities, and hematological indicators with CI-AKI incidence underscore the multifaceted nature of this phenomenon.

### Funding
This work was supported by CNPQ (National Council for Scientific and Technological Development) through the Universal Faixa A grant (Process: 404350/2023-2). The funders had no role in study design, data collection and analysis, decision to publish, or preparation of the manuscript.

### Grant Disclosures
The following grant information was disclosed by the authors:
CNPQ (National Council for Scientific and Technological Development) through the Universal Faixa A grant: 404350/2023-2.

### Competing Interests
The authors declare that there are no competing interests.

## Author Contributions

- Matheus Santos Moitinho conceived and designed the experiments, performed the experiments, analyzed the data, prepared figures and/or tables, authored or reviewed drafts of the article, and approved the final draft.
- Dulce Barbosa conceived and designed the experiments, performed the experiments, authored or reviewed drafts of the article, and approved the final draft.
- Attilio Galhardo conceived and designed the experiments, performed the experiments, prepared figures and/or tables, authored or reviewed drafts of the article, and approved the final draft.
- Adriano Caixeta conceived and designed the experiments, authored or reviewed drafts of the article, and approved the final draft.
- Eduesley Santana-Santos conceived and designed the experiments, authored or reviewed drafts of the article, and approved the final draft.
- Maximina Cunha analyzed the data, authored or reviewed drafts of the article, and approved the final draft.
- Beatriz Santana Prado analyzed the data, authored or reviewed drafts of the article, and approved the final draft.
- Cassiane Dezoti da Fonseca conceived and designed the experiments, performed the experiments, analyzed the data, authored or reviewed drafts of the article, and approved the final draft.

## Human Ethics

The following information was supplied relating to ethical approvals (i.e., approving body and any reference numbers):

The study was approved by the ethics and research committee under Comitê de Ética em Pesquisa da Universidade Federal de São Paulo (CEP-UNIFESP) opinion no. 3.763.447.

## Data Availability

The raw measurements are available in the Supplementary File.

## Supplemental Information

Supplemental information for this article can be found online at http://dx.doi.org/10.7717/peerj.19166#supplemental-information.

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
