# Peer review of "Mehran vs. Mehran2 pre-procedure: which score better predicts risk of contrast-induced acute kidney injury in patients with acute coronary syndrome?"

_PeerJ, doi:10.7717/peerj.19166_

## Round 0.1 · original submission · Major Revisions

Dear Dr. da Fonseca,

Your manuscript entitled " Mehran vs. Mehran2 pre-procedure: which score better predicts risk of contrast-induced acute kidney injury in patients with acute coronary syndrome?" which you submitted to PeerJ, has been reviewed by the editor and 2 experts in the field.

The reviewers generally support your work but have raised several significant concerns that must be addressed before the manuscript can move forward. In particular, please consider the following revisions:

• Language and Clarity: Simplify the language to improve overall clarity.
• Figures and Tables: Enhance the detail and consistency of your figures and tables.
• Patient Selection: Provide a stronger rationale for your patient selection criteria.
• Study Limitations: Acknowledge and discuss the limitations of your study.
• Statistical Evaluation: Broaden your statistical evaluation.
• Analysis of Findings: Deepen the analysis of your findings.

Addressing these issues will improve your work's readability and strengthen its scientific rigor and clinical relevance. I would happily reconsider your manuscript if you undertake these substantial revisions and resubmit.

If you decide to resubmit the revised version, please summarize all the improvements made in the new version and give answers to all critical points raised in the reviewers’ report in an accompanying letter. Copy and paste each and every reviewer's comment above your response. Please consider these points carefully, as the revised manuscript will undergo a second round of review by the same reviewers.

I hope you will be prepared to make the necessary amendments and submit a revised manuscript with a statement of how you responded to the reviewers’ comments.

Yours sincerely,
Stefano Menini

·

Basic reporting

See detailed comments below

Experimental design

See detailed comments below

Validity of the findings

see detailed comments below

Additional comments

In the index report, the authors explore the relationship and predictive ability of risk of contrast-induced acute kidney injury in patients with acute coronary syndrome using Mehran vs. Mehran2 scores. They demonstrated that The original Mehran score was more consistently discriminative for predicting CI-AKI risk in ACS patients undergoing PCI compared to the Mehran 2 Pre-procedural score. The analysis is timely, clinically relevant and of great use to physicians, and cardiologists. Tables and figures are used effectively to present complex data. The study employs robust statistical techniques, including ROC curve analysis and Poisson regression, to assess predictive capacity and validate findings. The comparison between scores is well-structured. Key metrics such as sensitivity, specificity, and AUC-ROC are systematically reported.

I have minor comments:-
1) The manuscript contains repetitive phrases and occasional complex sentence structures, which could hinder readability.
2) : Simplify sentences and eliminate redundancies. For example, replace "the performance of the Mehran 2 Pre-procedural score was slightly inferior" with "The Mehran 2 Pre-procedural score underperformed slightly."
3) The figures, such as the ROC curve, lack sufficient detail in their captions. Table labels are not consistently formatted, and legends can be improved for clarity.
4) Revise captions to be self-explanatory. Ensure consistent formatting and include details like "confidence intervals" or "statistical tests used."
5) The introduction provides an adequate background but could expand on the rationale for focusing on ACS patients exclusively.
6) Explain the clinical implications of excluding stable angina patients, particularly how this affects the broader applicability of findings.
7) The study population is relatively small and limited to a single center in Brazil, which may limit generalizability. Highlight this limitation more prominently and discuss its impact on external validity. Suggest multicenter studies for future research.
8) The study uses AUC-ROC as a key metric but does not explore other complementary measures of predictive power, such as reclassification indices. Include alternative metrics (e.g., net reclassification improvement) to provide a more comprehensive assessment of score performance.
9) The manuscript adequately explains most statistical tests but lacks details on model assumptions for Poisson regression. State how assumptions (e.g., no overdispersion) were tested and whether the Poisson model was compared with alternatives like negative binomial regression.
10) The study notes differences in findings compared to prior research but does not deeply analyze potential causes. Discuss these differences in the context of population characteristics, such as ethnicity or comorbidities, and their effect on score validity.
11) Highlight scenarios where the Mehran2 score's superior specificity could prevent unnecessary interventions.
12) Discuss the relevance of the original Mehran score's higher sensitivity for early detection in at-risk populations
13) Suggest a hybrid approach: using the original Mehran score for ACS patients at higher procedural risk and the Mehran2 score for broader, pre-procedural stratification.
14) Add a side-by-side comparison of score performance across patient subgroups to visually represent nuances in applicability

Reviewer 2 ·

Basic reporting

No comments, all comments are listed collectively at the bottom.

Experimental design

No comments, all comments are listed collectively at the bottom.

Validity of the findings

No comments, all comments are listed collectively at the bottom.

Additional comments

Dear author;
I reviewed the article entitled ‘Mehran vs. Mehran2 pre-procedure: which score better predicts risk of contrast-induced acute kidney injury in patients with acute coronary syndrome?’. I found the article very interesting and useful for our journal. However, I have some minor comments before the acceptance of article.
--minor comments;
1-There are some grammatical mistakes in the article, hence, I recommend the proof-reading

2- Was any power analysis performed when determining the number of patients? It is recommended to specify.

3-Is there any explanation for the short patient recruitment period? The power of the study could have been increased by recruiting more patients. It is recommended to clarify this issue.

---

## Round 0.2 · accepted · Accept

Dear Dr. Fonseca,

Thank you for submitting the revised version of your manuscript. After reviewing the changes by the Reviewers and myself, I am pleased to inform you that all the reviewers' comments have been adequately addressed. Therefore, your manuscript is ready for publication in PeerJ.

I thank all reviewers for their efforts in improving the manuscript and the authors' cooperation throughout the review process.

Sincerely yours,
Stefano Menini

·

Basic reporting

see comments below

Experimental design

see comments below

Validity of the findings

see comments below

Additional comments

The authors deserve credit for their meticulous approach to this revision. They have satisfactorily responded to all the concerns raised and the same is being reflected in the revised version. No further comments

Reviewer 2 ·

Basic reporting

Thank you for revision.

Experimental design

Thank you for revision.

Validity of the findings

Thank you for revision.

Additional comments

Thank you for revision.